# Volatilome Analysis in Prostate Cancer by Electronic Nose: A Pilot Monocentric Study

**DOI:** 10.3390/cancers14122927

**Published:** 2022-06-14

**Authors:** Alessio Filianoti, Manuela Costantini, Alfredo Maria Bove, Umberto Anceschi, Aldo Brassetti, Mariaconsiglia Ferriero, Riccardo Mastroianni, Leonardo Misuraca, Gabriele Tuderti, Gennaro Ciliberto, Giuseppe Simone

**Affiliations:** 1Department of Urology, IRCCS—“Regina Elena” National Cancer Institute, 00144 Rome, Italy; filianoti@gmail.com (A.F.); manuela.costantini@ifo.it (M.C.); alfredo.bove@ifo.it (A.M.B.); umberto.anceschi@ifo.it (U.A.); aldo.brassetti@ifo.it (A.B.); maria.ferriero@ifo.it (M.F.); riccardomastroianniroma@gmail.com (R.M.); leonardo.misuraca@ifo.it (L.M.); gabriele.tuderti@ifo.it (G.T.); 2Department of Urology, San Filippo Neri Hospital, 00135 Rome, Italy; 3Scientific Direction, “Regina Elena” National Cancer Institute, 00144 Rome, Italy; gennaro.ciliberto@ifo.it

**Keywords:** cancer screening, electronic nose, gas sensor array, prostate cancer, tumor biomarkers, volatilome

## Abstract

**Simple Summary:**

Although challenging and highly expensive for health systems worldwide, no useful markers are available in clinical practice that aim to anticipate prostate cancer diagnosis in the early stages in the context of wide population screening. Urine analysis via an electronic nose provides volatile organic compounds easily usable in the diagnosis of urological diseases. Some previous works suggested that dogs trained to smell urine could recognize lung, bladder, or breast cancer with various success rates, but no strong results have been published. Based on this, the present study tested the ability of urinary volatilome profiling to distinguish patients with prostate cancer from healthy controls, proving to be a promising, non-invasive diagnostic tool with high accuracy in discriminating patients from controls. Its ease of use and low costs make the findings widely reproducible, suggesting that in the future, there will be the possibility of reducing the number of invasive procedures such as prostate biopsies in clinical practice.

**Abstract:**

Urine analysis via an electronic nose provides volatile organic compounds easily usable in the diagnosis of urological diseases. Although challenging and highly expensive for health systems worldwide, no useful markers are available in clinical practice that aim to anticipate prostate cancer (PCa) diagnosis in the early stages in the context of wide population screening. Some previous works suggested that dogs trained to smell urine could recognize several types of cancers with various success rates. We hypothesized that urinary volatilome profiling may distinguish PCa patients from healthy controls. In this study, 272 individuals, 133 patients, and 139 healthy controls participated. Urine samples were collected, stabilized at 37 °C, and analyzed using a commercially available electronic nose (Cyranose C320). Statistical analysis of the sensor responses was performed off-line using principal component (PCA) analyses, discriminant analysis (CDA), and ROC curves. Principal components best discriminating groups were identified with univariable ANOVA analysis. groups were identified with univariable ANOVA analysis. Here, 110/133 and 123/139 cases were correctly identified in the PCa and healthy control cohorts, respectively (sensitivity 82.7%, specificity 88.5%; positive predictive value 87.3%, negative predictive value 84.2%). The Cross Validated Accuracy (CVA 85.3%, *p* < 0.001) was calculated. Using ROC analysis, the area under the curve was 0.9. Urine volatilome profiling via an electronic nose seems a promising non-invasive diagnostic tool.

## 1. Introduction

At present, prostate cancer (PCa) represents the second most frequent cancer among men and was the fifth leading cause of cancer death in 2020. Incidence rates for this condition range from 6.3 to 83.4 per 100,000 men across all regions, with higher rates found in some countries of North America, northern and western Europe, Australia/New Zealand, and Southern Africa, while lower rates are registered in North Africa and Asia. Although it is a very common disease, relatively little is still known about the etiology of prostate cancer.

The wide variation observed worldwide in prostate cancer incidence rates is mainly ascribed to the different diagnostic practices in the various countries mentioned.

In fact, in Europe, the USA, and Australia, following the widespread introduction of the prostate-specific antigen (PSA) test, which allowed the detection of preclinical cancers, there have been rapid increases in prostate cancer incidence rates since the late 1980s/early 1990s [1]. To date, the diagnosis and follow-up of PCa is well defined but has a high economic impact on the national health system.

Various metabolic and inflammatory pathways in the body are known to release thousands of volatile organic compounds (VOCs) into urine. Some of these volatile biomarkers can be associated with the presence of abnormal clinical conditions such as foreign bodies, infections due to bacteria, or other microorganisms, which are always linked to changes in the composition of VOCs. Urinary VOCs have been proposed as alternative biomarkers without finding, currently, a specific molecular pattern for prostate cancer. Numerous studies have highlighted more than 100 urinary VOCs, and some of these are always present in urine, mainly ketones from protein degradation. 

In the recent literature, different articles suggested that dogs trained to smell urine (sniffer-trained dogs) could recognize several types of cancer such as lung, bladder, or breast tumors with various success rates, but strong results have not yet been published [2,3].

In particular, some works have already proved that post specific olfactory training, dogs can recognize PCa from urine with promising rates of sensitivity and specificity.

The concern about dogs and their olfaction is the poor reliability and reproducibility on a large scale. 

The gas sensor array, also known as the electronic nose (eNose) might represent an encouraging diagnostic device for several medical settings, being a tool with advanced technology that is able to reproduce a dog’s sense of smell for the identification of volatile components in human biofluids [4]. The eNose is an advanced device composed of thin-film nanocomposite sensors that provide diverse chemical interactions, including reversible electron transfer reactions, for the measurement of gases, acids, bases, oxidizers, and many other compounds. When the tool is exposed to a sample, it generates a unique smell-print which, through a specific pattern recognition system, can be analyzed in order to investigate its nature and origin [5].

From this perspective, eNose appears to be a very well suited device for the qualitative analysis of complex gaseous molecular mixtures. Indeed, these devices, due to their intrinsic analytical capabilities, are commonly applied in food and agricultural quality control, odor monitoring, safety, environmental pollution, indoor air quality control, and chemical industries, and in some military applications [6,7,8,9,10,11]. To date, the expansion of this analytical technique has led to the use of the eNose in various medical applications, particularly in the analysis of breathed air for early cancer detection [12].

It is problematic to sample exhaled air because it requires both cooperation and technique from the patient along with immediate analysis, while it is more feasible to test urine in clinical practice because it is easier to obtain and store. Several metabolic studies have already studied urine using Gas/Liquid Chromatography–Mass Spectrometry (G/LC-MS). According to preliminary data, the detection of urological malignancies using urine headspace is possible.

eNoses can be considered an innovative tool for VOCs sampling, because these devices are able to identify a mixture of VOCs, converting them into a urine profile (urine smell-print). These devices ensure an immediate, easy, inexpensive, on-site distinction of urine smell-prints by pattern recognition, without currently being able to identify individual molecular components. Using the urine headspace analysis of VOCs, this study tested the hypothesis that patients with prostate cancer can be discriminated from healthy controls.

## 2. Materials and Methods

### 2.1. Study Cohort

A total of 272 individuals were enrolled in this prospective, observational study.

The study population was divided in 2 groups: 133 patients with a diagnosis of PCa (PCa group) and 139 participants acting as a healthy control group (HC group). Measurements were performed between December 2019 and December 2021. The members of the PCa group were defined by the presence of histological evidence of neoplasm diagnosed by prostate biopsy. Patients were radiologically staged using currently accepted consensus criteria and underwent surgery. Patients with clinically established conditions affecting the urinary VOCs spectrum were not eligible for participation, in particular, those with renal dysfunctions, hematuria, dysuria, UTIs, and lithiasis of the urinary tract. The control group was composed of 139 subjects, recruited among personal contacts and composed of people who were ambulatory patients undergoing a routine visit, with a negative history of urinary symptoms and without evidence of any known neoplastic disease. The study was approved by the “Regina Elena” National Cancer Institute Ethics Committee, and all patients gave their written informed consent.

### 2.2. Electronic Nose and Measurement Chamber

For this study, we tested a commercially available electronic nose (Cyranose 320, Smith Detections, Pasadena, CA, USA). It is a handheld device with a nano-composite array of 32 organic polymer sensors, and its operation is quite simple: as soon as the sensors are exposed to a mixture of VOCs, the polymers expand, stimulating a variation in their electrical resistance [13]. The system is able to record raw data as changes in the resistance of each of the 32 sensors in an on-board database, thus generating a distribution (urine smell-print) that details the VOC mixture and that can be applied in pattern-recognition algorithms. The measurement chamber consists of a 40 mL polystyrene vial and a Teflon soft cover top in which two holes are made. The first channel is for airflow, and the second was used with a modified 16G intravenous cannula, which was connected to provide sample air for the eNose. This modified cannula was recycled during the study and sterilized between measurements. The measurements were performed prior to surgery. Urine samples were collected in sterile conditions and sampled immediately using the eNose at a stable temperature of 37 °C in order to prevent the decay of metabolites, including VOCs. We collected 20 mL urine samples and maintained them at 37 °C in a water thermostatic bath, and they were subsequently analyzed. Each measurement lasted approximately 7 min, and a recovery period of 10 min was used to avoid carryover. The sampling was repeated to provide duplicate samples.

### 2.3. Data Analysis

Initial data were collected from eNose’s on-board database and analyzed with SPSS software (version 20.0, SPSS Inc., Chicago, IL, USA), performing analysis strategies that designedly restrict false discoveries [14]. Data were reduced to a set of principal components (PCA), obtaining the largest amount of variance in the original 32 sensors (two-dimensional principal component analysis (2D-PCA)). Then, univariate ANOVA analysis was applied to identify the principal components that were best differentiated among groups. Afterward, these principal components were considered to perform a linear canonical discriminant analysis (CDA) to collocate cases into a categorical partition. We adopted the “leave-one-out method” to compute the Cross Validated Accuracy percentage (CVA, %), an evaluation of accuracy of a predictive model in clinical practice. The CVA provides a percentage that assesses how precisely a predictive model will perform in practice. The probability of a positive diagnosis was estimated on basis of the CDA canonical discriminant function for each case. These probabilities were subsequently used to provide a receiver operator curve (ROC curve) with a 95% confidence limit.

## 3. Results

The baseline demographic and clinical features of the study cohorts are described in Table 1. The PCa and healthy control cohorts were homogeneous in terms of the baseline clinical data. The pathological stages of the PCa cohort are reported in Table 2. The best discriminating principal component groups were identified with univariable ANOVA analysis. Based on 2D-PCA, 110/133 and 123/139 cases were correctly identified in the PCa and healthy control cohorts, respectively (Figure 1). In the CDA, CVA was 85.3% (*p* < 0.001), sensitivity 82.7%, and specificity 88.5%; the positive predictive value was 87.3% and the negative predictive value 84.2% (Table 3). In the ROC analysis, the discrimination accuracy (area under the curve (AUC)) of the model was 0.90 (Figure 2). Repeated analysis of the independent second urine samples provided comparable findings (CVA 85%, AUC 0.9, sensitivity 83.1%, and specificity 87.6).

## 4. Discussion

The worldwide impact of oncologic diseases on public health is significant. One strategy to lower its burden is by pursuing cancer screening and early detection. If patients are diagnosed at an early stage of disease, their long-term overall survival and disease-free survival rates increase significantly, while related medical expenses decrease [15,16,17]. Despite the advances in technology, there is still a need for cheap diagnostic tests that are non-invasive to perform, sensitive, and reliable at the same time. Tissue biopsy is, to date, the most accurate diagnostic tool available for cancer detection, staging, and prognosis; however, its invasiveness represents a huge limitation. Often, it is challenging to obtain enough tissue for a reliable diagnosis, or else, the histology results are inconclusive. 

In this scenario, the liquid biopsy technique represents one of the new frontiers for cancer detection [18,19,20,21]. This technique is based on the identification of specific biomarkers in the circulating blood, in particular cell-free DNA (cfDNA) released in the blood stream from apoptotic or necrotic cells. The cfDNA released by neoplastic cells is called circulating tumor DNA (ctDNA). The tumor-specific mutations in ctDNA sequences targeted by a liquid biopsy allow for the identification of cancer patients from healthy controls [22,23]. The traditional tissue biopsy technique is far more invasive, less feasible, and requires specific expertise when compared to the liquid biopsy technique. Furthermore, since every tumor site releases specific ctDNA sequences into the bloodstream, a liquid biopsy allows for a more comprehensive tumor heterogeneity evaluation in comparison to the classic tissue biopsy [24]. This represents a crucial advantage, since the ctDNA plasmatic concentration has been proved to correlate with tumor size and stage [25,26]. Nonetheless, the potential advantages offered by the liquid biopsy technique in terms of disease detection, profiling, and treatment selection did not translate into a significat improvement in patients’ outcomes or medical costs when compared to the standard tissue biopsy, mainly due to the low concentration of ctDNA in the blood. Therefore, to date, only a few liquid-biopsy-based assays are actually available for clinical practice, while many others are still objects of investigations and debate. Consequently, other sources of biochemical information, besides ctDNA, are needed to increase liquid biopsy sensitivity and specificity. Cancerous cells’ metabolic pathways often lead to the formation of altered membrane proteins, capable of inducing cell membrane peroxidation and forming volatile organic compounds (VOCs) such as TMPRSS2 or PCA3 that can be detected in the headspace of cells [27]. Some of the VOCs produced by tumors are released into the atmosphere through breath, sweat, or in urine. Each and every organic compound has a distinctive odor, and even when present in minute quantities, they can be detected by dogs due to their exceptional olfactory acuity [28]. The hypothesis that dogs may be able to detect malignant tumors using their olfaction was first introduced in 1989 by Williams and Pembroke in a letter to *The Lancet* [29]. Since then, many studies have been conducted to explore the diagnostic power of trained dogs’ olfaction for lung, breast, bladder and prostate cancer detection, obtaining a wide discrepancy in the results. For example, in 2010, Cornu et al. first reported the efficacy of trained dogs’ olfaction on human urine samples in detecting PCa. In their experiment, sensitivity and specificity were both 91%. Nonetheless, the small sample size (with only 66 patients supplying urine and one dog) impaired the reliability of the results [30]. Similarly, in 2015, Taverna et al. reported 97% accuracy in the same context (PCa diagnosis) with two trained dogs [31]. In the setting of bladder cancer diagnosis, Willis et al. reported successful discrimination in 41% of the cancer cohort (95% CIs 23–58%) with six dogs [32]. Concerning the limitations of studies regarding trained dogs, the authors acknowledged kidney function, diet, or medications as potential olfactory misleading causes. Ultimately, all studies involving dogs present major drawbacks. First of all, to develop the discriminative ability, dogs require adequate and lengthy training by a professional team, which is costly and highly time-consuming. Secondly, the dog’s breed and the specific methodology used may influence the overall detection accuracy. Thirdly, dogs are not able to reliably work for a few hours consecutively. Due to the poor reliability, scalability, and reproducibility of the canine model on a large scale, the application of canine units in routine clinical practice is definitely inhibited. Notwithstanding, these studies shed light on the potential role of VOCs analysis in the clinical setting. Thus, other cheap and reproducible diagnostic tests using VOCs analysis are required by the medical community worldwide. A large spectrum of instrumentation is already available which allows for VOCs identification during pretty short time spans. The electronic nose is a portable, inexpensive, and easy-to-use diagnostic tool, able to perform the analysis of VOCs without the need for a specialized technician. In clinical practice, the eNose has already been utilized to discriminate bacterial cultures or to detect urinary tract infections, diabetes, or kidney diseases [33]. Urine is particularly rich in metabolites, and since it is so easy to harvest and available in large amounts, it is the preferred VOC source. Bernabei et al. tested the eNose in the differential diagnosis of urinary tract cancers (bladder and prostate cancer) from each other without attempting to distinguish a PCa cohort from healthy people [34]. Subsequently, preliminary feasibility studies that aimed to identify specific PCa urine smell-prints have been conducted, providing solid rationale to subsequent studies such as the present one. In 2021, Capelli et al. conducted a reproducibility study for eNose application in the PCa setting. They investigated the effect of different sample preparations on VOC’s analysis results, showing that the sample preparation technique, the urine volatiles’ extraction technique, and conditioning temperature affected the diagnostic performance of the eNose. 

The study involved fewer subjects than those included in our study (192 versus 272 cases), reporting slightly lower sensitivity and specificity rates [35]. In a study by Roine et al., based on the use of the ChemPro 100 eNose, which aimed to discriminate PCa from benign prostatic hyperplasia (BPH) patients, the sensitivity and specificity rates were 78% and 67%, respectively [36]. These data may be partially explained by the intrinsic risks of misclassifying BPH patients based on available diagnostic tests, because prostate biopsies, like all biopsy procedures, can give false negative cases. 

To the best of our knowledge, our study is the most extensive investigation into urinary volatilome profiling using an eNose in the field of PCa. In the CDA, we reported a CVA of 85.3%, with sensitivity and specificity equal to 82.7% and 88.5%, respectively. The high positive (87.3%) and negative predictive values (84.2%) make eNose a reliable option that deserves further validation as a promising screening tool in this specific clinical setting. In the ROC analysis, the discrimination accuracy of the model was 0.90. These findings were also confirmed by repeated analysis of the independent second urine sample set (CVA 85%, AUC 0.9). Interestingly, these results are consistent with those reported in the few previous studies conducted in this setting based on trained sniffer dogs, which reported a discrimination accuracy close to 90% [30,31]. 

Thanks to the high specificity achieved, the eNose represents a promising tool to address issues related to the current diagnostic procedures, such as patients’ overtreatment, complications associated with invasive procedures, and their management’s high costs for the national healthcare system. On the other hand, the eNose has proven reliable for anticipating PCa detection in the early stages in the context of wide population screening. A strength of this technology is certainly its low costs while combining benefits related to its non-invasive nature, both aspects that potentially impair wide reproducibility into clinical practice. The possibility of using volatile compounds from patients’ urine as a new PCa diagnostic tool seems fascinating; however, there are still several limitations to overcome before introducing the eNose as a routine diagnostic test in clinical practice. Firstly, volatilomes derived from exogenous sources, such as diet, medications, smoking, or alcohol consumption, can contaminate the sample to be tested. For this reason, endogenous VOCs’ selection and separation from exogenous volatilomes is a paramount step prior to the analysis of VOCs to avoid artifact contamination [37]. Secondly, the eNose cannot identify and quantify every single compound found in a sample; it can only detect specific molecular patterns. Furthermore, VOCs’ concentration patterns can vary substantially in different body fluids. This results in the identification of different sets of biomarkers in different body fluids, related to the same disease. Therefore, the paramount goal for future studies will be to identify as many volatile compounds as possible, all attributable to the same molecule which, in turn, can be identified as a novel marker for PCa diagnosis or a novel molecular target for systemic treatments. Specific limitations of our study include the lack of an external validation cohort and the single-center design of the study. Another consideration is that, surely, the results obtained in this pilot study in terms of sensitivity and specificity are lower than those of radiological examinations such as magnetic resonance. We believe that, today, MRI is the gold standard for prostate cancer detection, but it is expensive, time-consuming, and not easily reproducible [38]. The proposed diagnostic tool is not intended to replace the clinical use of MRI, which is able, among other things, to define which patients should be biopsied or not according to the PIRADS classification. 

## 5. Clinical Translation

We propose urinary VOCs analysis via eNose as a preliminary diagnostic tool due to its reproducibility, low cost, and ease of use. Despite the acknowledged limitations, the eNose provided diagnostic accuracy comparable to the PSA test (Appendix A). Thanks to its non-invasive nature, we believe that the eNose can be a potential tool for use as a fast mass screening test to be integrated with the use of a simple PSA test, of which the diagnostic accuracy in some cases is questionable [39]. 

In conclusion, we hypothesize the enhancement of the normal diagnostic iter for prostate cancer, proposing an easy-to-perform test that can integrate the use of PSA test more economically than magnetic resonance imaging and less invasively than prostate biopsies.

## 6. Conclusions

The electronic nose proved to be a valid tool with high potential in discriminating the urine smell-prints of patients with PCa from those of healthy controls, thus confirming trained sniffer-dogs experiments in the same PCa diagnostic setting. Its ease of use, combined with its non-invasive nature and low costs, makes the eNose a promising diagnostic clinical tool at least in the setting of PCa diagnosis, with the potential for wide reproducibility compared with initial reports based on trained animals.

## Figures and Tables

**Figure 1 cancers-14-02927-f001:**
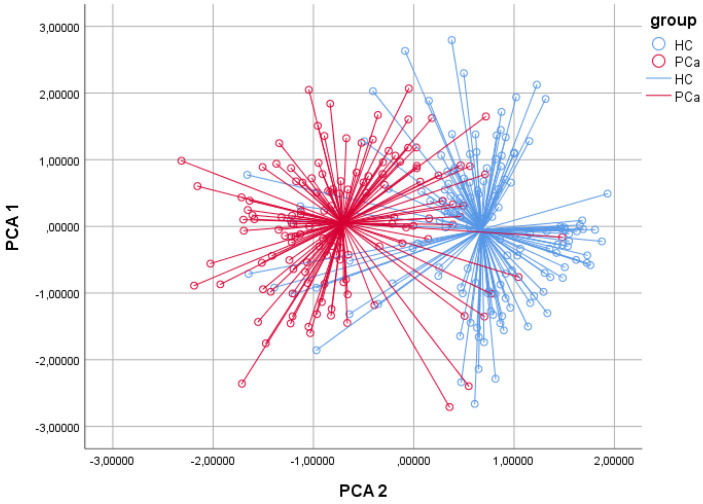
Two-dimensional principal component analysis plot. The two-dimensional PCA plot showed that patients with prostate cancer could be distinguished from healthy controls. Results obtained with CDA demonstrated correct classification in 85.3% of cases (*p* < 0.001). In ROC analysis, discrimination accuracy between PCa patients and healthy controls was 0.90. Repeated analysis using the second measure of each collected urine sample provided comparable findings. Sensitivity and specificity were 83.1% and 87.6%, respectively.

**Figure 2 cancers-14-02927-f002:**
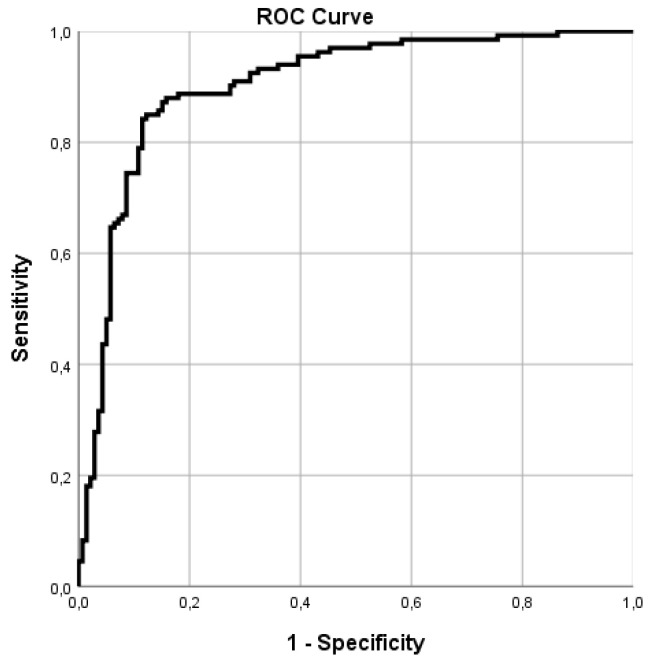
ROC curve analysis. In ROC analysis, discrimination accuracy (area under the curve (AUC)) was 0.9.

**Table 1 cancers-14-02927-t001:** Baseline characteristics of the study population.

Variable	PCa Group	HC Group	*p*-Value
N	133	139	
Age (years), mean ± SD (range)	67.37 ± 6.10 (46–82)	65.97 ± 12.99 (42–90)	0.259
Psa (ng/mL), mean ± SD (range)	12.65 ± 37.05 (0.5–425)	3.34 ± 4.64 (0.5–425)	0.004
Smokers, N (%)	27 (20.3%)	22 (15.8%)	0.339
Comorbidities, N (%)			
- Arterial hypertension	69 (51.8%)	60 (43.1%)	0.15
- History of AMI	4 (3%)	2 (1.4%)	0.38
- COPD	4 (3%)	4 (2.8%)	0.94
- Dyslipidemia	23 (17.2%)	27 (19.4%)	0.65

**Table 2 cancers-14-02927-t002:** Tumor classification after surgery.

Prostate Cancer Histopathological Results (TNM and Gleason Score)			
TNM Stage	*n* (%)	Gleason Score	*n* (%)
T2a	8 (6)	3 + 3	30 (22.6)
T2b	4 (3)	3 + 4	52 (39.1)
T2c	71 (53.4)	4 + 3	25 (18.8)
T3a	38 (28.6)	4 + 4	21 (15.8)
T3b	12 (9)	4 + 5	3 (2.2)
		5 + 4	2 (1.5)
Total	133 (100)		133 (100)

**Table 3 cancers-14-02927-t003:** Group classification.

		Expected Group Membership		
	Group	HC	PCa	Total
Count	HC	123	16	139
	PCa	23	110	133
%	HC	88.5	11.5	100
	PCa	17.3	82.7	100

Based on 2D-PCA, 110/133 and 123/139 cases were correctly identified in the PCa and healthy control cohorts, respectively. For PCa group, canonical discriminant analysis (CDA) showed a CVA% of 85.3 (*p* < 0.001) with SE of 82.7%, SP 88.5%, VPP 87.3%, and VPN 84.2%.

## Data Availability

All data generated or analyzed during this study are included in this published article. The supplementary materials can be downloaded at https://drive.google.com/file/d/180jagtS08LFimVOc-olOy1AZqZvNigk3/view?usp=sharing (accessedon 18 May 2022).

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
