# Peer review of "Volatilome Analysis in Prostate Cancer by Electronic Nose: A Pilot Monocentric Study"

_cancers, 2022, doi:10.3390/cancers14122927_

Round 1

Reviewer 1 Report

In the current study, Authors tested the ability of urinary volatilome profiling to distinguish 133 patients with prostate cancer from 139 healthy controls. 110/133 and 123/139 cases were correctly identified in the PCa and healthy control cohorts, respectively (sensitivity 82.7%, specificity 88.5%; positive predictive value 87.3% negative predictive value 84.2%). Cross Validated Accuracy (CVA 85.3%, p<0.001) was calculated. At ROC analysis, Area Under 36 Curve was 0.9.

Strengths of the study are the prospective nature, a real-life representation of low, intermediate and high-risk prostate cancer patients, a good statistical methodology, that this study is a novelty in the field and the eNose represents an attractive novel method to be associated with the other already existing to diagnose prostate cancer

Major comments

  • A major issue is represented by the %s of sensitivity, specificity and accuracy in general, which ideally should be higher than 90% to match the MRI. However, these %s overcome those of other diagnostic tools, particularly the biochemicals. Could the Authors improve the discussion section facing deeply these considerations?
  • The study lacks of subanalyses based on population characteristics, particularly after stratification regarding variables such as PSA, GS, or risk classes in general. Could the Authors produce these subanalyses?

Minor:

  • Please put % after raw number in Table 2

Author Response

Interesting work, showing the great possibilities of using the electronic nose for the diagnosis of prostate cancer from a liquid matrix, which is urine. The planned tests are correct and the obtained results are correctly interpreted. The authors are also aware of certain limitations in the use of the electronic nose for this type of research. I think the manuscript is worth undergoing further evaluation. Below are my other detailed comments:

line 72, Gas Sensor Array or electronic nose, please write if there is a difference between these names or if it is the name of the same device. Often in the literature on the subject, the term electronic nose or sensor matrix is ​​used interchangeably.

We thank the reviewer for the comments. The text has been modified in accordance with the revision.

line 75-76, electronic nose not only consists of electrochemical sensors, but also other types of sensors, please correct it.

We thank the reviewer for the comments. The text has been modified in accordance with the revision.

line 80-82, there are many areas in which the electronic nose has found application, e.g. odor monitoring, safety, environmental pollution, indoor air quality control, chemical industry below are some citations on this topic:

Monitoring of odour nuisance from landfill using electronic nose Chem. Eng. Trans, 40 (2014), pp. 85-90

Metal oxide sensor arrays for detection of explosives at sub-parts-per million concentration levels by the differential electronic nose, Sens. Actuators B Chem, 161 (2012), pp. 528-533

Application of electronic nose for industrial odors and gaseous emissions measurement and monitoring – an overview, Talanta, 144 (2015), pp. 329-340

A novel classifier ensemble for recognition of multiple indoor air contaminants by an electronic nose, Sens. Actuators B Chem, 207 (2014), pp. 67-74

Customized design of electronic noses placed on top of air-lift bioreactors for in situ monitoring the off-gas patterns Bioprocess Biosyst. Eng, 35 (2012), pp. 835-842.

We thank the reviewer for this clarification; these application areas have been added in the text and in the references.

line 254-263, the authors compared two groups of patients, it was rather predictable that with the electronic nose these two groups would be well discriminated against. The challenge would be to use several groups of patients with different neoplasms for research and then to demonstrate high sensitivity and specificity for the detection of prostate cancer. Please comment on this.

We thank the reviewer you for this criticism. Aware of this important aspect, we are currently testing the use of e-nose to discriminate prostate cancer from other urological cancers. In particular, new analyses are currently ongoing to test the sensitivity and specificity of the electronic nose in recognizing prostate cancer from kidney and bladder cancers. We hope that these data will be the object of future publications.

line 275-277, what VOCs can be expected in the gas phase when diagnosing urine with suspected prostate cancer? Do the authors have such knowledge on this topic?

We thank the reviewer for this comment. We currently do not have in-depth knowledge of this interesting aspect, but we have planned an experimental approach with spectrophotometric analyses to investigate the composition of urinary gas phase VOCs in prostate cancer. We hope that the characterization of these VOCs will be the object of a future publication.

Reviewer 2 Report

Interesting work, showing the great possibilities of using the electronic nose for the diagnosis of prostate cancer from a liquid matrix, which is urine. The planned tests are correct and the obtained results are correctly interpreted. The authors are also aware of certain limitations in the use of the electronic nose for this type of research. I think the manuscript is worth undergoing further evaluation. Below are my other detailed comments:

line 72, Gas Sensor Array or electronic nose, please write if there is a difference between these names or if it is the name of the same device. Often in the literature on the subject, the term electronic nose or sensor matrix is ​​used interchangeably.

line 75-76, electronic nose not only consists of electrochemical sensors, but also other types of sensors, please correct it.

line 80-82, there are many areas in which the electronic nose has found application, e.g. odor monitoring, safety, environmental pollution, indoor air quality control, chemical industry

below are some citations on this topic:

  1. Monitoring of odour nuisance from landfill using electronic nose

Chem. Eng. Trans, 40 (2014), pp. 85-90

  1. Metal oxide sensor arrays for detection of explosives at sub-parts-per million concentration levels by the differential electronic nose, Sens. Actuators B Chem, 161 (2012), pp. 528-533
  2. Application of electronic nose for industrial odors and gaseous emissions measurement and monitoring – an overview, Talanta, 144 (2015), pp. 329-340
  3. A novel classifier ensemble for recognition of multiple indoor air contaminants by an electronic nose, Sens. Actuators B Chem, 207 (2014), pp. 67-74
  4. Customized design of electronic noses placed on top of air-lift bioreactors for in situ monitoring the off-gas patterns Bioprocess Biosyst. Eng, 35 (2012), pp. 835-842.

line 254-263, the authors compared two groups of patients, it was rather predictable that with the electronic nose these two groups would be well discriminated against. The challenge would be to use several groups of patients with different neoplasms for research and then to demonstrate high sensitivity and specificity for the detection of prostate cancer. Please comment on this.

line 275-277, what VOCs can be expected in the gas phase when diagnosing urine with suspected prostate cancer? Do the authors have such knowledge on this topic?

Author Response

The title of this paper is interesting to readers, the content of the paper should be beneficial to readers in some extent. But there are some problems listed below should be tackled:

There are some grammatical errors/typos in the paper, eg., “because is more…” in line 88, “VOSs” in line 95, “37C” in line 127, “eNose proved to be…” in line 267 etc.

We thank the reviewer for the comments. The text has been modified in accordance with the revision.

In all tables of the paper, the decimal points are expressed by commas, which makes readers confused.

We thank the reviewer for this suggestion. We have eliminated commas in all tables and inserted points.

The third row of Table 1 is difficult to understand, for example, I cannot know what the “65, 79+-12 99(42-90)” means.

We thank the reviewer for this comment. The third row of Table 1 reports the mean age of patients and healthy controls expressed in years ± the standard deviation (SD). In brackets we reported the range with the minimum and maximum age found.

It will be better to give a comment for the meaning of the first column in Table 2.

In table 2 we have inserted an initial row with the explanation of the columns.

It seems that the title of Table 2 is contradictory with the sentence in line 124 “The measurements were performed prior to surgery”

The measurements were performed post prostate biopsy to be sure that the subjects belonged to experimental group were correctly inserted in prostate-cancer group, but prior to surgery. So, the table 2 shows the definitive histological classification obtained post-surgery (radical prostatectomy).

As we know, PCA, particularly 2d-PCA, will lose some information of the samples, which may also be the reason of low score of sensitivity/specificity in the paper. Why don’t the authors use more components such as 3d, or even 4d-PCA to improve the performance of the classifier?

We thank the reviewer for this comment. In this pilot study we have extracted 4 PCA. Variance percentage was: PCA1 84.966% – PCA2 10.124% - PCA3 1.338% – PCA4 0.864%. we used only 2d-PCA (using PCA1 and PCA2) because using the others PCA will not improve performance of the classifier.

Most topics of Section 4 (Discussion) should belong to Section 1 (Introduction).

We thank the reviewer for this comment. In reality, the text of the introduction and discussion was structured with the intention of correctly interpreting and respecting the policy of the Cancer journal which required:

The introduction should briefly place the study in a broad context and highlight why it is important. It should define the purpose of the work and its significance… As far as possible, please keep the introduction comprehensible to scientists outside your particular field of research”.

“Authors should discuss the results and how they can be interpreted from the perspective of previous studies and of the working hypotheses. The findings and their implications should be discussed in the broadest context possible”.

The authors said e-nose is proved to be reliable in early stage PCa detection/screening (see line 267-268), but no strong evidences were provided in the paper. The authors should explain the sensitivity/specificity for different stages of prostate cancer, because in early stage, the concentration of biomarker VOC should be very low, which may not be in the LoD (Limit of Detection) range of Cyranose 320.

We thank the reviewer for the comment. At this time, we don’t have the adequate patients cohort to perform stratification based on stage of disease.  We will reserve a sub-analysis or stratification based on risk class once an adequate patient cohort has been obtained. So we have provided to modify properly the sentence.

Typical/representative plots of response of Cyranose 320 should be given.

We thank the reviewer for the comment. The representative graphic plots of response (scattered dot graphic) is available on figure 1.

In line 321, it is said that all data generated or analyzed are included in this published article, but I didn’t find it. Besides, my suggestion is that all the data, code, flowchart of this paper be provided by supplementary manner or other way so as to improve the persuasiveness of the paper and to make it more valuable to readers.

The supplementary materials can be download at https://drive.google.com/file/d/180jagtS08LFimVOc-olOy1AZqZvNigk3/view?usp=sharing

Reviewer 3 Report

The title of this paper is interesting to readers, the content of the paper should be beneficial to readers in some extent. But there are some problems listed below should be tackled:

  • There are some grammatical errors/typos in the paper, eg., “because is more…” in line 88, “VOSs” in line 95, “37C” in line 127, “eNose proved to be…” in line 267 etc.
  • In all tables of the paper, the decimal points are expressed by commas, which makes readers confused.
  • The third row of Table 1 is difficult to understand, for example, I cannot know what the “65, 79+-12 99(42-90)” means.
  • It will be better to give a comment for the meaning of the first column in Table 2.
  • It seems that the title of Table 2 is contradictory with the sentence in line 124 “The measurements were performed prior to surgery”
  • As we know, PCA, particularly 2d-PCA, will lose some information of the samples, which may also be the reason of low score of sensitivity/specificity in the paper. Why don’t the authors use more components such as 3d, or even 4d-PCA to improve the performance of the classifier?
  • Most topics of Section 4 (Discussion) should belong to Section 1 (Introduction).
  • The authors said e-nose is proved to be reliable in early stage PCa detection/screening (see line 267-268), but no strong evidences were provided in the paper. The authors should explain the sensitivity/specificity for different stages of prostate cancer, because in early stage, the concentration of biomarker VOC should be very low, which may not be in the LoD (Limit of Detection) range of Cyranose 320.
  • Typical/representative plots of response of Cyranose 320 should be given,
  • In line 321, it is said that all data generated or analyzed are included in this published article, but I didn’t find it. Besides, my suggestion is that all the data, code, flowchart of this paper be provided by supplementary manner or other way so as to improve the persuasiveness of the paper and to make it more valuable to readers.

Author Response

In the current study, Authors tested the ability of urinary volatilome profiling to distinguish 133 patients with prostate cancer from 139 healthy controls. 110/133 and 123/139 cases were correctly identified in the PCa and healthy control cohorts, respectively (sensitivity 82.7%, specificity 88.5%; positive predictive value 87.3% negative predictive value 84.2%). Cross Validated Accuracy (CVA 85.3%, p<0.001) was calculated. At ROC analysis, Area Under 36 Curve was 0.9.

Strengths of the study are the prospective nature, a real-life representation of low, intermediate and high-risk prostate cancer patients, a good statistical methodology, that this study is a novelty in the field and the eNose represents an attractive novel method to be associated with the other already existing to diagnose prostate cancer.

Major comments

A major issue is represented by the %s of sensitivity, specificity and accuracy in general, which ideally should be higher than 90% to match the MRI. However, these %s overcome those of other diagnostic tools, particularly the biochemicals. Could the Authors improve the discussion section facing deeply these considerations?

We thank the reviewer for this comment. Surely , the results obtained in this pilot study in terms of sensitivity and specificity are lower than those of radiological examinations such as magnetic resonance. We totally agree that, at today, MRI is the gold standard for prostate cancer detection, but it is expensive, time-consuming and not easily reproducible. The proposed diagnostic tool is not intended to replace the clinical use of MRI, which is able, among other things, to define which patients should be biopsied or not according to the PIRADS classification. We propose e-nose as a preliminary diagnostic tool due to its reproducibility, less cost and easy to use (compared to MRI). Despite some reported limitations, the aim of this study was to enhance the normal diagnostic iter, proposing a non-invasive and fast test to perform mass screening in the population more accurately than using the simple PSA, of which the diagnostic accuracy is questionable.

We added these considerations in the Discussion section.

The study lacks of subanalyses based on population characteristics, particularly after stratification regarding variables such as PSA, GS, or risk classes in general. Could the Authors produce these subanalyses?

We thank the reviewer for the comment. This is a monocentric pilot study that aim the possibility of identifying prostate cancer patients in the general population. We will reserve a sub-analysis or stratification based on risk class once an adequate patient cohort has been obtained.

Minor:

Please put % after raw number in Table 2

We thank the reviewer for the comment. In table 2 we have inserted the % value for each number reported.

Round 2

Reviewer 1 Report

Authors answered my comments

Author Response

We thank the reviewer for their thoughtful comments and suggestions.

Reviewer 2 Report

I accept the revised manuscript and recommend it for further steps. Please correct the citations, as citations 7-11 do not contain the names of the authors.

Author Response

We thank the reviewer for this comment. We have corrected the 7-11 references as indicated.

In attachment you can find the changes made (in red)
